

# Retention of indoxyl sulfate in different genotypes of *ABCC2* may explain variation in tacrolimus pharmacokinetics

Jing Wang[1,*], Siqi Huang[2,*], Yuanchen Li[3], Qiu Fang[2], Min Wang[1] and Huaijun Zhu[1]

[1] Department of Pharmacy, Nanjing Drum Tower Hospital the Affiliated Hospital of Nanjing University Medical School, Nanjing, Jiangsu, China
[2] Department of Pharmacy, Nanjing Drum Tower Hospital Clinical College of Nanjing University of Chinese Medicine, Nanjing, Jiangsu, China
[3] Department of Pharmacy, China Pharmaceutical University Nanjing Drum Tower Hospital, Nanjing, Jiangsu, China
* These authors contributed equally to this work.

Corresponding authors
Min Wang,
wangmin19881013@126.com
Huaijun Zhu,
huaijun.zhu@gmail.com

## ABSTRACT

**Background:** Microbiota-derived toxins indoxyl sulfate and hippuric acid were previously reported to be associated with altered pharmacokinetics of the immunosuppressant tacrolimus in liver transplant recipients, and ABC transporter proteins are likely to be involved in the transport of such substances, but the *in vivo* role has not been elucidated. The aim of this study was to assess the retention of indoxyl sulfate and hippuric acid in the plasma of liver transplantation subjects carrying different genotypes of *ABCB1* and *ABCC2* (changes in transporter activity due to genetic variation), and to explore whether genetic variation is involved in altering the relationship between microbe-derived toxins and tacrolimus pharmacokinetics.

**Methods:** Liver transplantation subjects treated with the immunosuppressive regimen tacrolimus, corticosteroids, and mycophyolate mofetil were included and divided into normal renal function group and chronic kidney disease group. The plasma concentrations of indoxyl sulfate and hippuric acid in two groups of liver transplantation subjects carrying different genotypes of *ABCB1* and *ABCC2* were compared. For genotype carriers with significant differences, the Pearson Correlation Coefficient method was further used to investigate the correlation between plasma indoxyl sulfate level and tacrolimus dose-corrected trough concentration in patients with different renal function status.

**Results:** Carriers of the rs717620-24T variant exhibited high plasma indoxyl sulfate retention in patients with normal renal function, and furthermore, chronic kidney disease patients and patients with normal renal function exhibited indoxyl sulfate and tacrolimus in the *ABCC2* normal function ($\beta = -0.740$, $p = 0.020$) and reduced function groups ($\beta = -0.526$, $p = 0.005$), respectively, showing a strong correlation with tacrolimus.

**Conclusion:** *ABCC2* may be one of the pathways by which tacrolimus pharmacokinetics is altered by indoxyl sulfate.

# INTRODUCTION

Since tacrolimus (TAC) entered clinical use in the 1990s, it has rapidly become the cornerstone of immunosuppressive maintenance in solid-organ transplantation. Large intra- and inter-individual variability in the pharmacokinetics (PK) of TAC has been reported, necessitating precise therapeutic drug concentration monitoring (*Brunet et al., 2019*). Researchers have worked to find predictors of TAC PK, with demographics (*Brazeau et al., 2020*; *Tornatore et al., 2022*), drug-drug interactions (*Leino et al., 2019*; *Lin et al., 2015*; *Zhao et al., 2022*), and genetic factors (*Chauhan et al., 2023*; *Su et al., 2019*) being among the more clear-cut influences on TAC concentrations, in addition to studies suggesting that microbiota-derived toxins (*Degraeve et al., 2024*; *Lee et al., 2015*; *Manes et al., 2023*) contribute to alterations in TAC PK.

As byproducts of the gut microbiota's metabolism of dietary nutrients, microbiota-derived toxins have been linked to a number of illnesses, such as kidney disease (*Wu et al., 2011*), cardiovascular disease (*Stubbs et al., 2016*), inflammatory states (*Lin, 2019*; *Weingarden & Vaughn, 2017*). These toxins accumulate in a state of renal insufficiency. Indoxyl sulfate (IS, one of microbiota-derived toxins) was demonstrated to regulate hepatic P-gp (*ABCB1*) *via* AhR in rodent and cell culture models of chronic kidney disease (CKD). Clinical studies revealed that increased P-gp expression in CKD characterized by high plasma levels of IS may result in increased hepatic metabolism of cyclosporine (P-gp substrate) (*Santana Machado et al., 2018*). *Degraeve et al. (2023)* similarly found that regulation of *ABCB1* expression by intestinal bacterial metabolites leads to changes in TAC PK. *Guo et al., (2024)* observed that IS upregulates the expression of *ABCB1*, *ABCC2* in peripheral blood mononuclear cells from patients with CKD. Our previous study also found a correlation between plasma concentrations of microbiota-derived toxins IS and Hippuric acid (HA) and dose-corrected trough concentrations of TAC in liver transplant recipients (*Zhu et al., 2022*). Based on the current results, efflux transporters appear to be a pathway by which microbiota-derived toxins affect TAC PK, and CKD patients have a greater probability of being affected.

*ABCB1* and *ABCC2* are members of the family of ATP binding cassette (ABC) transporters. These are plasma membrane transporters that are expressed in various organs such as liver, intestine, kidney and brain (*Dietrich, Geier & Oude Elferink, 2003*), mainly involved in the absorption of TAC in the intestinal. At the same time, these efflux pumps are also involved in the transport of endogenous metabolites. Well-known examples are the Dubin–Johnson syndromes (*Corpechot et al., 2020*; *Moriondo et al., 2009*), where patients have elevated conjugated bilirubinemia caused by mutations affecting MRP2 (*ABCC2*). The expression and activity of these transporter proteins may differ in individuals due to genetic polymorphisms or pathological conditions (*Hirouchi et al., 2004*; *Laechelt et al., 2011*; *Wolking et al., 2015*). This will lead to individual

differences in the bioavailability of different drugs, toxins and carcinogens (of food origin). Single nucleotide polymorphisms (SNPs) in genes encoding efflux transporter proteins show potential to alter plasma exposure to both exogenous and endogenous substances (*Bruckmueller & Cascorbi, 2021*; *Haufroid, 2011*; *Muhrez et al., 2017*; *Soria-Chacartegui et al., 2021*). Therefore, it is necessary to complement studies on the effect of SNP-related alterations in transporter activity on the *in vivo* retention of microbiota-derived toxins.

Our study population was liver transplant recipients, who have a certain probability of evolving chronic kidney disease, which further leads to the accumulation of microbiota-derived toxins in the body and increases the probability of toxin-drug interactions. The aim was to assess the impact of known functional variants of *ABCB1* and *ABCC2* on the retention of indoxyl sulfate or hippuric acid in the plasma of liver transplant recipients, and whether they are associated with TAC PK variability under different states of renal function.

## MATERIALS AND METHODS

### Study design and participants

In the current study, we enrolled 124 patients who were hospitalized at Nanjing Drum Hospital from June 2022 to April 2023. All recruited patients underwent triple-therapy immunosuppression protocol consisting of TAC, corticosteroids, and mycophenolate mofetil.

Inclusion criteria were the following: (1) age at least 18 years; (2) patients experiencing liver transplantation for the first time; (3) liver transplant 12 months previously, treatment with a calcineurin inhibitor (TAC, Prograf®, Astellas, Killorglin, Ireland; or Saifukai®, Huadong Medicine, Hangzhou, China) and available calcineurin inhibitor assay data. Patients who were being co-treated with potent CYP3A4/5 or P-gp inducers and inhibitors including ketoconazole, itraconazole, wuzhi capsule et al. were excluded. Patients with combined organ transplantations were also excluded. Blood samples were analyzed routinely in the pharmacology department in order to determine TAC trough concentrations during the follow-up of liver transplant patient. For each patient, residual whole blood samples were divided into two sub-samples: one for the detection of IS and HA concentrations and one for genotyping.

This study was approved by the ethics committees of the Nanjing Drum Tower Hospital (No. 2020-053-01). The scientific aims and procedures regarding this study were explained to each patient by means of a telephone informed consent and their verbal consent was obtained.

### Data collection

The patient's demographic and laboratory data (sex, age, body weight, height, serum creatinine, total bilirubin, Alanine transaminase et al.) were extracted from electronic medical records. Body Mass Index (BMI) was calculated from weight (kg) divided by square of height ($m^2$). Used the Modification of Diet in Renal Disease equation to calculate the eGFR. CKD stages were evaluated according to the Kidney Disease: Improving Global Outcomes Classification (*Cockwell & Fisher, 2020*).

## Materials and determination of plasma levels

HA was purchased from Sigma Aldrich (Saint-Quentin Fallavier, France), IS was purchased from Toronto Research Chemicals (North York, Canada), 13C6-IS, d5-HA were purchased from isoreag (Shanghai, China). Ultrapure water was produced in laboratory by a Milli-Q Reagent Water System from Millipore (Darmstadt, Germany).

Blood samples for TAC monitoring were collected before morning administration. Blood trough concentration of TAC was detected by EMII 2000 TAC assay kit (Siemens Healthcare Diagnostics Inc., Erlangen, Germany) with a microparticle enzyme immunoassay (Viva-E, Siemens Healthcare Diagnostics Inc, Erlangen, Germany), the linear range of the assay was 2.5–30 ng/mL. Column temperature was $24 \pm 2$ °C, and the detection wavelength was 340 nm.

Plasma levels of toxins (IS and HA) were determined using liquid chromatography-tandem mass spectrometry. Briefly, toxins were extracted from 20 μL of plasma by adding 80 μL of an ice-cold methanol solution containing internal standards ($^{13}C_6$-IS, $d_5$-HA), briefly vortexed for 3 min, following centrifugation (10 min at 18,800 $g$), 50 μL of supernatant was transferred to a new micro-centrifuge tube, and then reconstituted with 50 μL of ultrapure water and vortexed for 1 min, centrifuge at 18,800 $g$ for 10 min. The compounds were separated on a ACQUITY UPLC HSS T3 column (1.8 μm, 2.1 × 100 mm; Waters, Milford, MA, USA) using a gradient of acetonitrile (B) and ultrapure water with 10 mM ammonium acetate (A) and a flow rate of 0.3 mL/min. The initial mobile phase consisted 10%B and 90%A. Gradient variation was as follows: 10%–90%B at 0–5 min, and, thereafter, to 90%B in 10s, the post run time was 3 min. The injection volume was 10 μL with a needle wash between injections by 50% methanol aqueous solution.

Data were acquired in multiple reaction monitoring mode after negative-mode electrospray ionization. Monitored ion transitions were m/z 211.8 → 80.0 for IS (collision energy = −29.1 V), m/z 217.5 → 80.0 for $^{13}C_6$-IS (collision energy = −28.7 V), m/z 177.9 → 134.1 for HA (collision energy = −14.9 V), m/z 183.0 → 139.2 for d5-HA acid (collision energy = −16.1 V).

## Genotyping

Five SNP genotypes of the two genes ABCB1 and ABCC2 in liver transplant recipients were determined by Sanger sequencing. Includes four SNPs (rs2032582 (G2677, exons 21)), rs1045642 (C3435T, exons 26), rs717620 (−24C>T, exon 1), rs3740066 (c.3972C>T, exons 28) associated with decreased substrate transporter activity, and one SNP (rs2273697 (c.1249G>A, exons 10)) that increases substrate transporter activity (*Hirouchi et al., 2004*; *Laechelt et al., 2011*; *Wolking et al., 2015*).

Briefly, total genomic DNA was extracted from blood using the Ezup Column Blood Genomic DNA Purification Kit (Sangon Biotech, Shanghai, China) according to the manufacturer's instructions. Following quality control, the DNA fragments were amplified and sequenced by Sangon Biotech Co. (Shanghai, China) using an automated ABI 3730XL DNA Analyzer (Applied Biosystems, Foster City, CA, USA).

## Statistical analysis

The skewness and kurtosis tests were used to access whether continuous variables conform to normal distribution. Continuous variables with normal distribution are quoted as the mean (standard deviation) and Student's $t$-test was used to compare variables between the two groups. Continuous variables with a non-normal distribution are quoted as the median ([interquartile range (IQR)]), and the Wilcoxon rank sum test was used to compare. Categorical variables are quoted as the percentage and frequencies, and compared using the chi-squared test. The Hardy-Weinberg equilibrium test was performed using chi-squared test. The association between the genetic polymorphisms and the plasma concentration of microbiota-derived toxins were assessed in dominant genetic models. Multiple logistic regression models (dominant) to analyze the correlation between SNP data and microbial derived toxins. The C/D of TAC was computed as a pharmacokinetic index by dividing the trough by the daily dose (D). After evaluating the distribution using different transformation methods, the concentrations of toxins were transformed using the natural logarithm to achieve a more normal distribution. For statistical analyses, the concentrations of toxins/TAC below LOD were estimated as half of the LOD, with statistical significance set at $p < 0.05$. Linear regression analysis was used to study the correlation between toxin concentrations in plasma and tacrolimus PK. The Pearson Correlation Coefficient was used to establish the strength of the linear correlations (significance level set at $\alpha = 0.05$). All statistical analyses were performed using Stata 17.0.

# RESULTS

## Patients' characteristics and polymorphisms

A total of 124 patients were analyzed in this study, most of whom were Han Chinese from Jiangsu and Anhui provinces, with overall patients' characteristics described in Table 1. The median (interquartile range (IQR)) age of the participants was 55 (47–63) years, 62.9% were men, and the median body mass index (BMI) was 22.7 (20.1–24.3) kg/m$^2$. A small number of participants had diabetes (9.7%) or hypertension (8.9%). The group of patients with CKD was significantly older when compared with the group with normal renal function. When considering the laboratory data, patients with CKD had higher plasma levels of uric acid, and microbiota-derived toxins.

The minor allele frequency (MAF) of *ABCB1* rs2032582, rs1045642, and *ABCC2* rs717620, rs2273697, rs3740066 in patients with normal renal function were 38.78%, 35.71%, 17.35%, 13.27%, 19.39%, respectively. The minor allele frequency (MAF) of *ABCB1* rs2032582, rs1045642, and *ABCC2* rs717620, rs2273697, rs3740066 in patients with CKD were 34%, 37.33%, 19.33%, 12%, 18.24%, respectively. The distribution of all genotypes complied with the Hardy-Weinberg equilibrium, with the exception of *ABCB1* rs2032582 and *ABCC2* rs2273697 (Table 2).

## Plasma of microbiota-derived toxins

A summary of the concentrations (μg/mL) of the two microbiota-derived toxins detected in all populations is presented in Table 1. In all individual samples, total levels of all detected IS ranged from 0.04–5.53 μg/mL in plasma, while levels of HA in plasma ranged

**Table 1 Baseline characteristics of the study population.**

| Characteristics | Total | Patients | | p-value |
|---|---|---|---|---|
| | | CKD (n = 75) | Normal (n = 49) | |
| Age, year | 55 (47–63) | 58 (51–65) | 50 (39–57) | **<0.001** |
| Gender, male (%) | 78 (62.9) | 48 (64.0) | 30 (61.2) | 0.754 |
| BMI, kg/m$^2$ | 22.7 (20.1–24.3) | 22.1 (19.8–24.8) | 23 (20.8–24.2) | 0.666 |
| POD, day | 1,321 (829–1,654) | 1,319 (824–1,669) | 1,329 (880–1,610) | 0.880 |
| Diabetes, n (%) | 12 (9.7) | 12 (16.0) | 0 (0.0) | **0.003** |
| Hypertension, n (%) | 11 (8.9) | 11 (14.7) | 0 (0.0) | **0.005** |
| ALT, IU/L | 16.2 (12.1–28.1) | 15.3 (10.7–25.1) | 17.9 (14.7–32.8) | **0.025** |
| AST, IU/L | 21.4 (17.5–27.3) | 20.9 (16.7–26.0) | 22.8 (18.8–29.1) | 0.110 |
| TBIL, μmol/L | 14.5 (10.8–19.4) | 14.1 (10.1–19.3) | 15.1 (11.6–19.6) | 0.277 |
| Albumin, g/L | 44.2 (42.1–46.2) | 44.5 (42.1–46.3) | 43.9 (42.0–45.6) | 0.746 |
| Uric acid, μmol/L | 381.5 (323–451.5) | 401 (346–471) | 354 (312–416) | **0.015** |
| C-reactive protein, mg/L | 2.6 (1.9–3.8) | 2.6 (1.9–3.7) | 2.5 (1.9–3.9) | 0.832 |
| Creatine, μmol/L | 81.5 (66–100) | 96 (85–115) | 61 (55–70) | **<0.001** |
| eGFR, mL/min/1.73 m$^2$ | 83.3 (64.3–110.2) | 67.8 (52.5–78.7) | 114.7 (105.5–128.4) | **<0.001** |
| TAC concentration, ng/mL | 5.25 (3.80–6.40) | 5.40 (3.90–6.60) | 5.10 (3.60–6.20) | 0.105 |
| TAC dose, mg/day | 4 (2–4) | 4 (2–4) | 4 (2–4) | 0.658 |
| TAC concentration (ng/mL)/dose (mg/day) | 1.53 (1.14–2.40) | 1.55 (1.18–2.40) | 1.50 (1.08–1.95) | 0.393 |
| IS, μg/mL | 0.76 (0.43–1.4) | 1.00 (0.63–1.55) | 0.42 (0.25–0.82) | **<0.001** |
| HA, μg/mL | 0.18 (0.07–0.47) | 0.25 (0.10–0.57) | 0.13 (0.04–0.31) | **0.015** |

**Note:**
ALT, alanine transaminase; AST, aspartate aminotransferase; HA, hippuric acid; IS, indoxyl sulfate; POD, post of transplantation day; TBIL, total bilirubin. $p < 0.05$ is highlighted in bold.

from 0.005–4.21 μg/mL, showing large individual variability. Pearson's coefficient reveals good linear relationship between two toxins and eGFR ($\beta_{IS} = -0.013$, $p_{IS} < 0.001$; $\beta_{HA} = -0.007$, $p_{HA} = 0.045$, Fig. 1).

Figures 2 and 3 shows that individual concentrations for IS and HA in plasma samples, stratified by the *ABCB1* and *ABCC2* genotype of the participant. Patients with CKD exhibit higher microbiota-derived toxins concentrations relative to those with normal renal function.

## Association of *ABCC2* rs717620 with plasma retention of IS

The behaviour of HA and IS (increased or decreased) according to the genotypic class for each SNP is shown in Figs. 2 and 3. Plasma IS concentrations were lower in subjects with normal renal function in the CC genotype of *ABCC2* rs717620 than in carriers of the CT plus TT genotype, compared to subjects with CKD. However, we did not observe significant intergroup differences for HA.

Compared the characteristics of demographic and laboratory variables between the two groups of carriers of the *ABCC2* rs717620 CC genotype and the CT plus TT genotype, the results of univariate and multiple analysis are shown in Table 3, none of them appeared to be statistically different from each other.

**Table 2 SNPs of livers transplant patients and patients with different renal function included in the study.**

| Patients | SNP | Gene | Allele | MAF (%) | Genotype frequency | | | HWE-$p$ |
|---|---|---|---|---|---|---|---|---|
| | | | | | AA | AB | BB | |
| Total | rs2032582 | ABCB1 | T | 21.77 | 31 | 47 | 46 | 0.001 |
| | rs1045642 | ABCB1 | T | 36.69 | 50 | 57 | 17 | 0.906 |
| | rs717620 | ABCC2 | T | 18.55 | 80 | 42 | 2 | 0.243 |
| | rs2273697 | ABCC2 | A | 12.50 | 97 | 23 | 4 | 0.095 |
| | rs3740066 | ABCC2 | T | 18.70 | 80 | 40 | 3 | 0.564 |
| Non-CKD | rs2032582 | ABCB1 | T | 38.78 | 6 | 26 | 17 | 0.149 |
| | rs1045642 | ABCB1 | T | 35.71 | 19 | 25 | 5 | 0.437 |
| | rs717620 | ABCC2 | T | 17.35 | 32 | 17 | 0 | 0.320 |
| | rs2273697 | ABCC2 | A | 13.27 | 13 | 36 | 0 | 0.001 |
| | rs3740066 | ABCC2 | T | 19.39 | 31 | 17 | 1 | 0.666 |
| CKD | rs2032582 | ABCB1 | T | 34 | 7 | 37 | 31 | 0.001 |
| | rs1045642 | ABCB1 | T | 37.33 | 31 | 32 | 12 | 0.445 |
| | rs717620 | ABCC2 | T | 19.33 | 48 | 25 | 2 | 0.724 |
| | rs2273697 | ABCC2 | A | 12.00 | 4 | 10 | 61 | 0.009 |
| | rs3740066 | ABCC2 | T | 18.24 | 2 | 23 | 49 | 1.000 |

Note:
Allele, minor allele code; HWE, Hardy–Weinberg equilibrium; MAF, minor allele frequency.

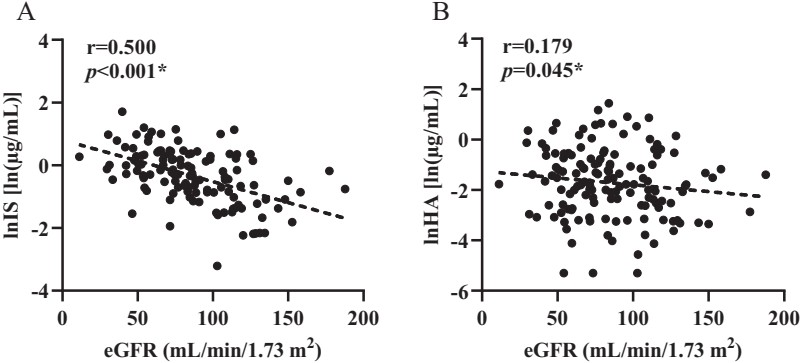

**Figure 1 Control correlations observed between estimated glomerular filtration rate and plasma levels of toxins.** (A) indoxyl sulfate (IS), (B) hippuric acid (HA). $p$ and $r$ values were obtained from Spearman correlational analyses ($n$ = 124). *$p < 0.05$.

## Correlation between plasma IS and TAC dose-corrected trough concentrations

Pearson correlation coefficient ($r$) was used to investigate the strength in the linear relationship among level of IS and TAC dose-corrected trough concentrations in groups with different renal functional status, for individuals with rs717620 CC, CT plus TT genotypes. As shown in Fig. 4, TAC dose-corrected concentrations exhibited a statistically significant negative correlation with IS plasma levels in both the CKD population harboring the CC genotype (Fig. 4A, β = −0.74, $r$ = 0.113, $p$ = 0.020) as well as the renal

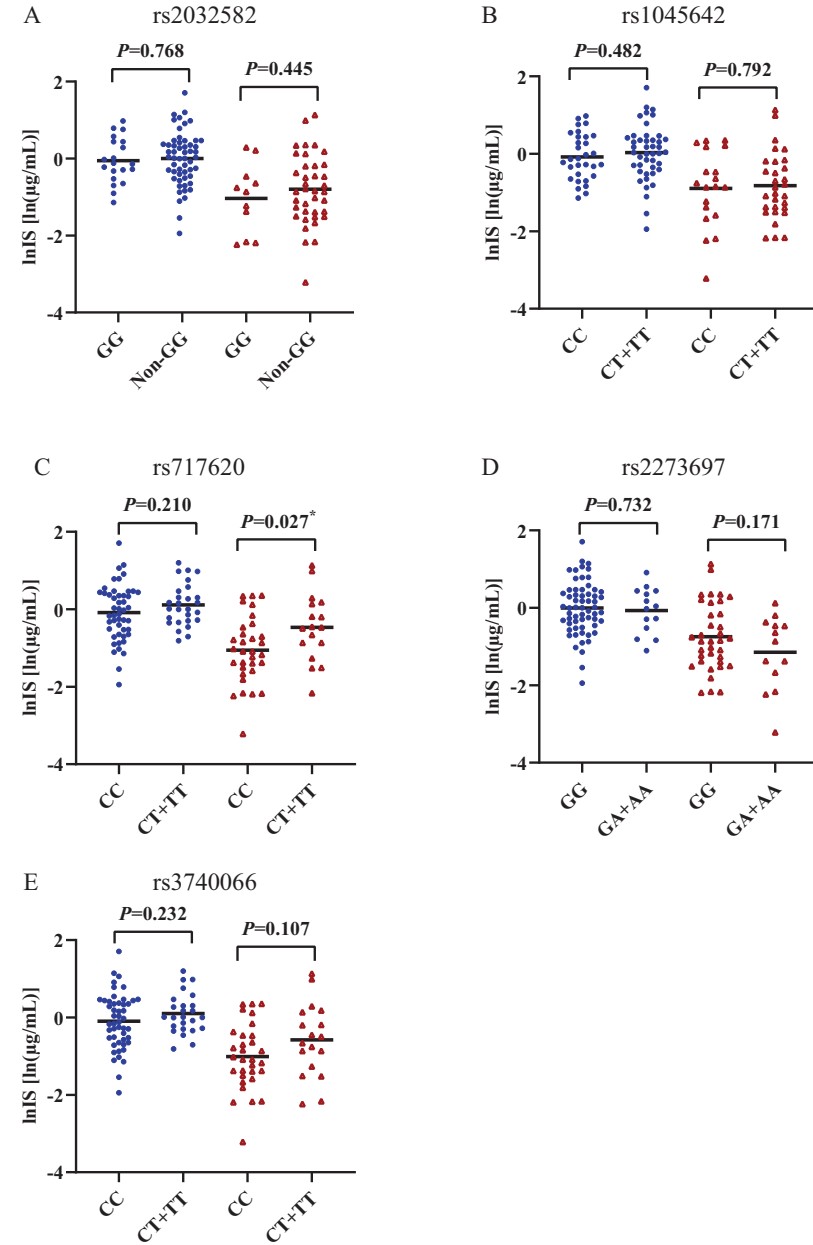

**Figure 2 Scatters plots of indoxyl sulfate in patients with CKD and normal renal function, stratified by the *ABCB1* and *ABCC2* genotype of the participant.** (A) *ABCB1* rs2032582, (B) *ABCB1* rs1045642, (C) *ABCC2* rs717620, (D) *ABCC2* rs2273697, (E) *ABCC2* rs3740066. Non-GG represents GA+GT+AT +TT+AA. *$p < 0.05$. The blue origin represents patients with CKD and the red triangle represents patients with normal renal function.

function normal population harboring CT plus TT genotype (Fig. 4D, β = −0.526, $r = 0.415$, $p = 0.005$). However, no such correlation was found in the other two groups.

## DISCUSSION

We examined the concentrations of two microbiota-derived toxins, IS and HA, in the plasma of 124 liver transplant recipients by liquid chromatography tandem mass

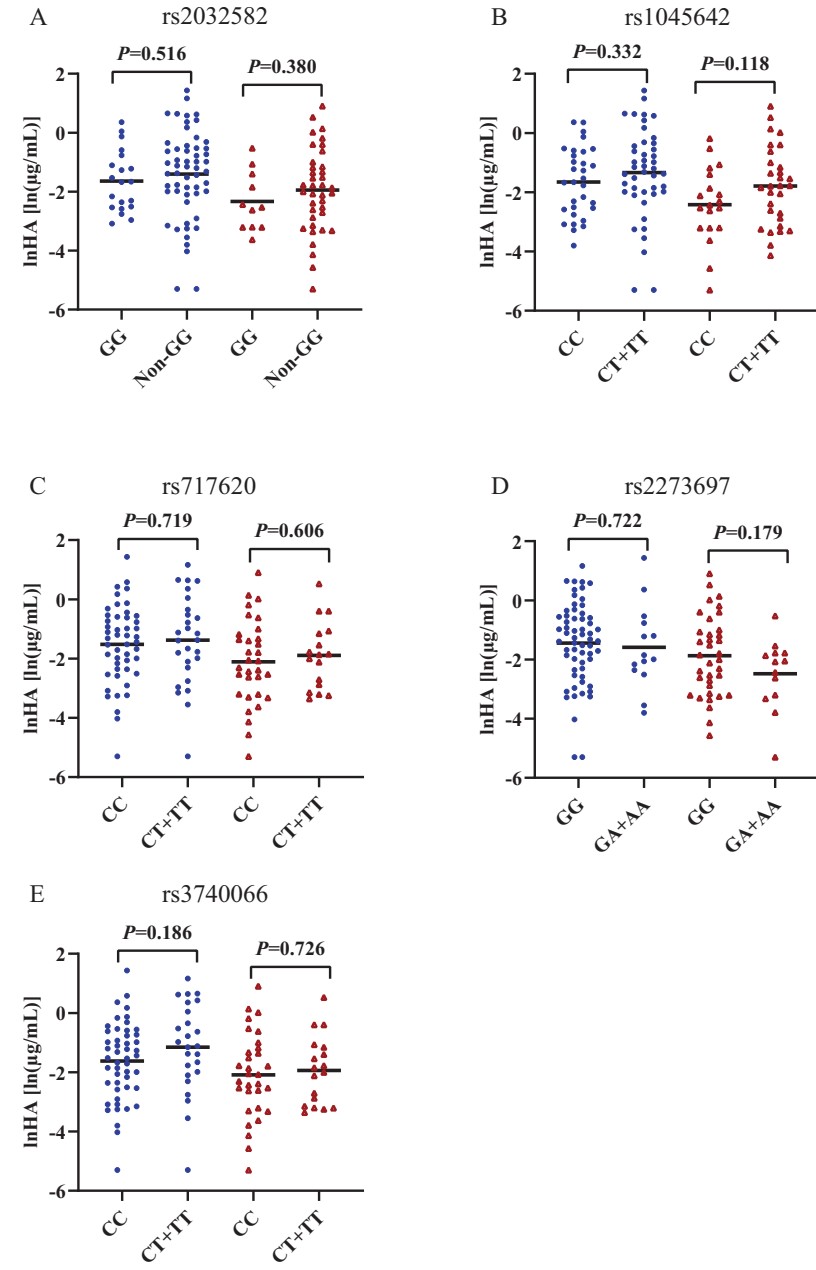

**Figure 3 Scatters plots of hippuric acid in patients with CKD and normal renal function, stratified by the *ABCB1* and *ABCC2* genotype of the participant.** (A) *ABCB1* rs2032582, (B) *ABCB1* rs1045642, (C) *ABCC2* rs717620, (D) *ABCC2* rs2273697, (E) *ABCC2* rs 3740066. Non-GG represents GA+GT+AT +TT+AA. The blue origin represents patients with CKD and the red triangle represents patients with normal renal function.

spectrometry, and compared the concentrations of microbiota-derived toxins in the plasma of subjects harboring different genotypes according to five SNP genotypes of the two genes ABCB1 and ABCC2. This study sought to elucidate the relationship between genetic polymorphisms in drug transporters, microbiota-derived toxins, and TAC pharmacokinetics in a population of liver transplant recipients. The present results shown

**Table 3 Univariate and multiple logistic regression analysis of factors associated with the *ABCC2* rs717620 genotype (*n* = 124).**

| Characteristics | Univariate analysis | | Multiple analysis | |
|---|---|---|---|---|
| | OR [95%CI] | *p*-value | OR [95%CI] | *p*-value |
| Age, year | 0.024 [−0.009 to 0.058] | 0.159 | | |
| Sex, male | −0.102 [−0.102 to 0.387] | 0.792 | | |
| BMI, kg/m$^2$ | −0.068 [−0.189 to 0.054] | 0.277 | | |
| POD, day | <0.001 [−0.001 to 0.001] | 0.426 | | |
| Diabetes | −0.105 [−1.366 to 1.156] | 0.870 | | |
| Hypertension | 0.458 [−0.790 to 1.706] | 0.472 | | |
| ALT, IU/L | 0.003 [−0.019 to 0.024] | 0.814 | | |
| AST, IU/L | −0.011 [−0.037 to 0.015] | 0.404 | | |
| TBIL, μmol/L | −0.036 [−0.086 to 0.013] | 0.146 | | |
| Albumin, g/L | −0.019 [−0.089 to 0.052] | 0.605 | | |
| Uric acid, μmol/L | <0.001 [−0.003 to 0.003] | 0.950 | | |
| C-reactive protein, mg/L | 0.012 [−0.034 to 0.058] | 0.604 | | |
| Creatine, μmol/L | 0.007 [−0.002 to 0.016] | 0.120 | | |
| eGFR, mL/min/1.73 m$^2$ | −0.007 [−0.020 to 0.004] | 0.191 | | |
| lnIS [ln(μg/mL)] | 0.527 [0.060 to 0.994] | 0.027 | 0.527 [0.060 to 0.994] | 0.027 |

**Note:**

ALT, alanine transaminase; AST, aspartate aminotransferase; IS, indoxyl sulfate; POD, post of transplantation day; TBIL, total bilirubin.

that liver transplant recipients carrying different genotypes of *ABCC2* rs717620 exhibit different plasma IS retention, which may correlate with the expected function of the studied SNPs. We also found that CKD subjects carrying the rs717620 CC genotype, as well as subjects with normal renal function carrying the rs717620 CT+TT genotype, had plasma IS levels associated with changes in TAC PK. Our results support the possibility that *ABCC2* may be the pathway by which the microbiota-derived toxins IS alters TAC PK. To the best of our knowledge, this is the first assessment of the association of microbiota-derived toxins (IS and HA) with transporter activity involving genetic variation, TAC PK variation, and our findings provide a corresponding addition to the *in vivo* linkage of the three.

The common *ABCC2*-24C>T (rs717620) variant, located in the 5′ untranslated region (5′ UTR). Regarding the impact of genetic factors on *ABCC2* activity, *in vitro* experiments by *Laechelt et al. (2011)* showed that the −24T variant was associated with reduced *ABCC2* (MRP2) protein expression. Research shows that this polymorphism causes an 18.7% decrease in protein activity of *ABCC2* in HepG2 cells (*Haenisch et al., 2008*). It also causes a decrease in mRNA expression in renal tissue in those who have it (*Haenisch et al., 2007*). *Chen et al. (2022)* also proposed that this polymorphism causes changes in the secondary structure of the mRNA, which ultimately affects inter-individual differences in clopidogrel. In the present study, IS concentrations tended to be higher in variant allele carriers for subjects with normal renal function who carried the rs717620-24T variant.

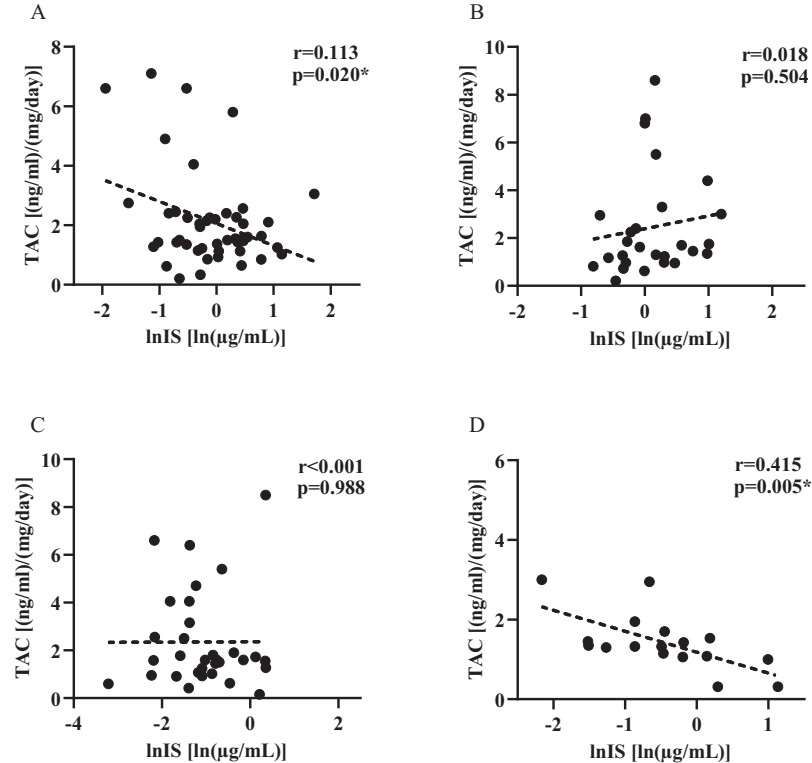

**Figure 4 Plasma indoxyl sulfate level and tacrolimus dose-corrected trough concentrations in groups with different renal functional status, for individuals with rs717620 CC, CT plus TT genotypes.** (A) CKD patients who were rs717620 CC genotype carriers, (B) CKD patients who were rs717620 CT plus TT genotype carriers, (C) normal renal function patients who were rs717620 CC genotype carriers, (D) normal renal function patients who were rs717620 CT plus TT genotype carriers. The correlation was analyzed by Spearman rank correlation coefficient, dotted line represents regression line. *$p < 0.05$.

Indeed, we considered that between diminished- and enhanced-function *ABCC2* gene polymorphisms, if plasma levels of an endogenous metabolite change in an opposite manner, this metabolite is likely to be an endogenous substrate of *ABCC2* (*Muhrez et al., 2017*). For example, through nontargeted metabolite profiling of human urine samples, *Song et al. (2012)* identified metabolites that are excreted differently based on the genotype SLC22A2/OCT2. They next conducted experiments to establish that tryptophan was a particular endogenous substrate of OCT2 and that its renal clearance was lowered due to a diminished transport function linked to the SLCA22A2 variation. In the present study, individuals carrying the rs717620 CT plus TT genotype, which is considered to be dysfunctional for *ABCC2*, exhibited higher concentrations of IS compared to the functionally normal CC group, and we therefore conclude that IS is likely to be an endogenous substrate for *ABCC2*.

In HA, however, we do not find this feature, which can be explained by the chemical structure of the phenomenon. The two toxins differ structurally in that IS has an indole ring that is structurally similar to a series of aromatic derivative substances found in previous studies to functionally interact with *ABCC2 in vitro* (*Sjöstedt et al., 2017*), whereas

HA does not, a difference that may contribute to the specificity of the gene binding pattern. Therefore, *ABCC2* also promotes the excretion of organic anions conjugated with glutathione, glucuronate, or sulfate (*Dietrich et al., 2001*), and IS mostly present in the body as sulfate. In addition, *ABCB1* rs2032582 and *ABCC2* rs2273697 were not in Hardy-Weinberg equilibrium, perhaps because this was a single-center study and the number of genotypes detected was relatively small. Further research is still needed to explain this phenomenon.

For patients with normal renal function, carriers of the rs717620 CT+TT genotype showed a strong negative correlation between IS plasma levels and TAC PK (Fig. 4D, $p = 0.005$), whereas the 24T variant, which reduced the transporter activity of *ABCC2*, showed high levels of IS plasma retention, and we hypothesized that there might be some causal relationship between the three. It was recently shown that erythromycin is a substrate for human MRP2. Erythromycin is another macrolide lactone with comparable physicochemical properties to TAC (molecular weight, LogP, topological polar surface area) (*Franke et al., 2011*; *Hariharan et al., 2009*). The possibility exists that TAC competes with IS for the same site on the *ABCC2* transporter protein, however explaining this mechanism of action is beyond the scope of this study.

Although IS showed weak differences in CKD subjects carrying different genotypes of rs717620, the results were not statistically different. In addition, CKD patients harboring the CC genotype demonstrated a negative correlation between IS plasma levels and TAC PK (Fig. 4A, $p = 0.020$). Differences in observations between the two groups of subjects may reflect the main and secondary effects of genetic factors and renal function status. Since the transport of microbiota-derived toxins *in vivo* may also involve organic anion-transporting polypeptides (OATPs) (*Lowenstein & Nigam, 2021*; *Nigam & Granados, 2023*), the clearance of endogenous substances by OATPs decreases with decreasing renal function, which likewise alters the *vivo* levels of IS. In addition, when the effect of rs717620 gene stratification was removed, it was found that study subjects in CKD 2 exhibited a more significant correlation between TAC and IS levels ($β = −0.124$, $p = 0.017$, Fig. S1), whereas CKD patients harboring the CC genotype also had an eGFR in the range of CKD 2. Thus, there may be some coincidence in the association between IS and TAC levels observed in such patients. In conclusion, the correlation between genotyping based on *ABCC2* transporter activity and *in vivo* retention levels of IS in the state of CKD, as well as the correlation between IS levels and TAC PK variants, may be masked by factors such as OATP.

There are some limitations of this study, mainly because we only determined the genotypes of liver transplant recipients and lacked information on the genotypes of donors. However, TAC is an oral drug, and studying the permeability of intestinal transporter proteins to TAC can help to understand the process by which TAC enters the circulation. ABC transporter proteins are also highly expressed in the intestinal, and the recipient's genotype largely reflects the viability of intestinal ABC transporter proteins. In addition, numerous factors influence the intestinal microbiota, making it challenging to accurately assess the relationship between IS, HA, and TAC pharmacokinetics. Furthermore, the cross-sectional study design limits our ability to evaluate TAC exposure

over time and draw causal inferences. Despite this, our study provides new insights into the pharmacokinetic mechanisms of tacrolimus, with a focus on genetic factors and microbiota-derived toxins, which will guide the direction of our future research.

## CONCLUSIONS

Our study suggests that known functional variants of the *ABCC2* transporter protein alter the retention of microbiota-derived toxins in plasma and that this change may be associated with TAC PK variants. These findings provided evidence for a role of genetic factors. Considering that this is a cross-sectional study, more longitudinal studies are needed to confirm our conclusions.

## ACKNOWLEDGEMENTS

We sincerely thank the Clinical Pharmacology Division of the Department of Pharmacology at Nanjing Drum Tower Hospital for their outstanding support.

### Funding

This project was funded by the National Natural Science Foundation of China (NSFC81302849), the Nanjing Medical Science and Technique Development Foundation (No. YKK17075), and the Jiangsu Province Youth Medical Talents Project (No. QNRC2016013). The funders had no role in study design, data collection and analysis, decision to publish, or preparation of the manuscript.

### Grant Disclosures

The following grant information was disclosed by the authors:
National Natural Science Foundation of China: NSFC81302849.
Nanjing Medical Science and Technique Development Foundation: YKK17075.
Jiangsu Province Youth Medical Talents Project: QNRC2016013.

### Competing Interests

The authors declare that they have no competing interests.

### Author Contributions

- Jing Wang analyzed the data, prepared figures and/or tables, and approved the final draft.
- Siqi Huang performed the experiments, prepared figures and/or tables, and approved the final draft.
- Yuanchen Li analyzed the data, prepared figures and/or tables, and approved the final draft.
- Qiu Fang analyzed the data, prepared figures and/or tables, and approved the final draft.
- Min Wang conceived and designed the experiments, analyzed the data, authored or reviewed drafts of the article, and approved the final draft.
- Huaijun Zhu conceived and designed the experiments, authored or reviewed drafts of the article, and approved the final draft.
## Human Ethics

The following information was supplied relating to ethical approvals (*i.e.*, approving body and any reference numbers):

The ethics committees of the Nanjing Drum Tower Hospital granted Ethical approval to carry out this study.

## Data Availability

The raw data are available in the Supplemental File.

## Supplemental Information

Supplemental information for this article can be found online at http://dx.doi.org/10.7717/peerj.18729#supplemental-information.

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
