# Peer review of "Retention of indoxyl sulfate in different genotypes of ABCC2 may explain variation in tacrolimus pharmacokinetics"

_PeerJ, doi:10.7717/peerj.18729_

## Round 0.1 · original submission · Major Revisions

I recommend authors to follow the STREGA guidelines on reporting a genetic association study (PMID: 19278015)

·

Basic reporting

Dear editors,
The authors are congratulated on their efforts, which have not gone unnoticed. However, their work does have several limitations that need to be addressed.
The study design needs to be more appropriate to assess the interesting hypothesis. The idea is that intestinal microbiota can modulate ABCB expression through IA and HA production, thus modifying TAC exposure.
However, a more appropriate design would be best and should be implemented to assess this. First, a longitudinal study is much more suited to testing the hypothesis than cross-sectional sampling. The intestinal microbiota is not constant and. may be primarily influenced by diet, metabolites, and drugs. This information has to be captured to truly assess IS and HA. Furthermore, information on immunosuppression drug exposure should be detailed concerning TAC exposure over time and MMF. This latter drug shows enterohepatic circulation, which can be affected by intestinal microbiota and impact IA and HA. Finally, CKD (which needs definition) may be the consequence of TAC overexposure rather than efflux transporter activity.
I believe the hypothesis is worth investigating through a longitudinal study with even a limited number of patients whose ABCB genotype is defined a priori and even over a limited period if time.

Experimental design

Poor and unfit for testing the hypothesis. My comments are reported above.

Validity of the findings

To be questioned.

Reviewer 2 ·

Basic reporting

Lines 86-87 – I understand the authors' intent, but I find the justification for why SNPs might influence the transport of endogenous metabolites simply because they affect the transport of exogenous ones to be weak, especially due to the use of the word "logically." Why would this be logical? I suggest rephrasing the text so that sentences 83-86 and 86-87 are better integrated.
Line 53 – Is “Christians et al., 2002” the most recent reference on demographic predictors for tacrolimus pharmacokinetics?
Lines 91-99 – The paragraph starting with “We examined the concentrations of two...” seems inappropriate for an introduction as it details the methodology. I suggest simplifying what the authors did in the study in the introduction, linking it with the objectives, and leaving these details for the methods section or the beginning of the results discussion.
Additional comments on the introduction:
Overall, it is well-written and contains the necessary elements for an introduction to a genetic association study, such as biological plausibility for the choice of candidate genes and a well-described problem. However, the text could be more fluid. Some paragraphs seem disconnected from each other. I suggest better linking the text to increase its fluency.

Experimental design

General Comment
The methodology is well-explained and easily replicable. The authors have taken care to make the method simple and comprehensible. However, some adjustments are necessary.
Materials and Methods
Data Collection
Lines 106-107 – Are all the patients included in the study native to the Nanjing region? It is crucial to specify the ethnicity of the patients in the methodology. When not possible, at least the origin or the majority origin of the patients in the sample should be stated. This information is important for future studies that will cite this work, as ancestry cannot be inferred based solely on the hospital where the patients are hospitalized.
Genotyping
Line 168 – The authors should ensure that quality control is well-detailed to guarantee the accuracy of the results. Was any concordance testing performed for genotyping (were some random samples re-sequenced to confirm the results)?
Statistical Analysis
The statistical analysis methodology seems robust, using appropriate tests for different types of data and analyses (such as chi-square tests, Student’s t-tests, and multiple logistic regression).
The authors should address the assumptions of the tests and include in the text whether these were met before choosing the tests.
Lines 170-171 – The authors should verify whether the logarithmic transformation of toxin concentrations for normalization is appropriate for all data or if other statistical approaches might be more suitable. Were normality tests performed for the data? If so, which test?
Overall, the analysis methodology appears solid, but some areas could be more detailed.

Validity of the findings

Results
Overall, the results presented are clear and well-structured, but I suggest a few modifications.
Lines 186-189 – The Hardy-Weinberg equilibrium analysis was performed for the total sample of 124 transplant patients. However, the study involves a case-control setup with patients having normal and altered renal functions. The correct approach would be to conduct the equilibrium analysis for the two groups (based on renal function) separately.
Table 1 – Highlight significant p-values.
Table 2 – There is no issue with keeping the equilibrium test values for the total sample, but it would be useful to include the test separated by groups.

The analysis of demographic and laboratory characteristics between genotypic groups could be more detailed, especially to better understand if there are any confounding factors not considered.
Figures
The resolutions of the images in the figures are good. I suggest highlighting only significant p-values but keeping all values.
Scatter plots do not present p-values. The images could be standardized to include the test values, significant or not.
The legends are well-explained.
Discussion
The discussion is solid and addresses the main points of the study in detail. However, for a more significant impact, it would be beneficial to include a more in-depth analysis of the mechanism of action, clarify the interaction between renal function and genotype, and explore the clinical implications of the findings more extensively.
Conclusion
The conclusion effectively summarizes the main results and highlights the relevance of the findings for the field of pharmacogenomics and toxicology. However, I suggest adding a brief mention of the study's limitations.

Additional comments

The study makes a significant contribution to understanding the interaction between genetics, microbiota, and TAC pharmacokinetics. The methodology is solid, and the results are well-discussed, providing new insights into how genetic variants can influence toxin retention and TAC pharmacokinetics. However, the fluency of the text, especially in the introduction, could be improved, and more details about the methodology could be reviewed to strengthen the study's validity. The discussion of limitations and the clinical application of the findings could also be more detailed. Overall, the study is a valuable addition to the existing literature and provides a foundation for future research in the field.

Annotated reviews are not available for download in order to protect the identity of reviewers who chose to remain anonymous.

Reviewer 3 ·

Basic reporting

The article is interesting and well-written, but it requires clarification on a few points.

Experimental design

In the experimental section regarding determining plasma concentration, it is necessary to note the temperature and the detection wavelength.

Add the p-values or correlation coefficient r to the figures to facilitate the read

Validity of the findings

no comment

Additional comments

none

---

## Round 0.2 · accepted · Accept

You have addressed all the comments.

Reviewer 2 ·

Basic reporting

The authors have made almost all the suggested modifications. When not implemented, my question was satisfactorily addressed. I am pleased with the response to my feedback.

Experimental design

The modifications regarding the experimental design have been made.

Validity of the findings

The authors revised part of the discussion and conclusion of the paper, including the study's limitations and the impact of the results.

Additional comments

After the modifications made, I consider the manuscript ready for publication.